# Mechanisms of Fluoride Toxicity: From Enzymes to Underlying Integrative Networks

**Anna Strunecka * and Otakar Strunecky**

The Institute of Technology and Business, Okružní 517/10, 370 01 České Budějovice, Czech Republic;
otakar.strunecky@gmail.com
* Correspondence: anna.strunecka@gmail.com; Tel.: +420-608371116

**Abstract:** Fluoride has been employed in laboratory investigations since the early 20th century. These studies opened the understanding of fluoride interventions to fundamental biological processes. Millions of people living in endemic fluorosis areas suffer from various pathological disturbances. The practice of community water fluoridation used prophylactically against dental caries increased concern of adverse fluoride effects. We assessed the publications on fluoride toxicity until June 2020. We present evidence that fluoride is an enzymatic poison, inducing oxidative stress, hormonal disruptions, and neurotoxicity. Fluoride in synergy with aluminum acts as a false signal in G protein cascades of hormonal and neuronal regulations in much lower concentrations than fluoride acting alone. Our review shows the impact of fluoride on human health. We suggest focusing the research on fluoride toxicity to the underlying integrative networks. Ignorance of the pluripotent toxic effects of fluoride might contribute to unexpected epidemics in the future.

**Keywords:** autism spectrum disorders; aluminofluoride complexes; enzymes; fluoride toxicity; G proteins; IQ deficits; magnesium; neurotoxicity; phosphate

## 1. Introduction

More than 500 million people live in endemic fluorosis areas with an elevated level of fluoride in drinking water and biosphere with public health problems [1,2]. The fluorosis symptoms span from mild effects on teeth enamel, headaches, dizziness, loss of appetite, to severe pathological disturbances. These include dental and skeletal fluorosis, hypothyroidism, sleep disorders, inflammations, IQ deficits, and suspected autism [3–6].

In 1931, H. Trendley Dean, head of the Dental Hygiene Unit at the National Institute of Health (NIH), began investigating the epidemiology of fluorosis in the USA. Dean and his staff determined the fluoride levels in drinking water causing fluorosis. They discovered that fluoride levels of up to one milligram fluoride per liter of drinking water (one ppm) did not cause dental fluorosis in most people. Dean's reports formed the foundation of the concept that the ingestion of fluoride will augment the teeth enamel and make it less susceptible to dental caries [7]. Dean suggested adding fluoride to water sources deficient in fluoride to bring its concentration up to the optimal value. After trials in the USA, the World Health Organization (WHO) recommended community water fluoridation (CWF), and many countries implemented this program [8–10]. The CWF recommendation has been followed for over 70 years in developed countries, such as the USA, Canada, Australia, New Zealand, Ireland, and some parts of the United Kingdom (UK) [9].

CWF started an era of increased income from fluoride as never before in human history. Recently, 370 million people from 27 countries have a supply of fluoridated drinking water [9]. The number of fluoride sources has further expanded since fluoride is used for the improvement of food, beverages, hygiene, and medical products including fluorinated drugs.

The fluorine atom does not exist in nature in a free and unmixed state. The mineral fluorite ($CaF_2$) is the main source of fluorine for commercial use. Fluorine readily reacts with other elements to form fluoride compounds, within which it always adopts an oxidation state of −1. Inorganic compounds are formed with hydrogen, metals, and non-metals. Therefore, fluoride refers to the negatively charged ion of the fluorine atom. In general, soil fluoride is not available to plants. There are exceptions such as tea plants that are natural accumulators of fluoride from fertilizers [1,2,4]. Fruits such as blueberry and grape concentrate fluoride from soil and it might appear in commercial juices. Fluorine excess had been found to be detrimental also to plants. Injury to plants is common all around industries making aluminum, fertilizers, or glass. Special plants might produce the toxic organofluorine compound fluoroacetate, which is used as a defense against grazing by herbivores.

Conflicting views occur in debates addressing whether fluorine is an essential element for humans. The researchers of the Panel of the European Food Safety Authority (EFSA) state that fluoride has no known essential function in human physiology and development [11]. On the other hand, the WHO and the Centers for Disease Control and Prevention (CDC) consider fluoride to be an important dietary element for humans because of "resistance to dental caries is a physiologically important function" [12].

Recently, concern about whether fluoride supplemented via drinking water and all other sources does not have harmful effects remains widely discussed by the scientific community. The goal of our review is to enhance understanding of the mechanisms by which fluoride impacts human health and development.

## 2. Methods

We performed in-depth research on fluoride toxicity in humans from the available peer-reviewed articles until June 2020 utilizing numerous medical sites like PubMed, Scopus, and other internet sources. including the journal Fluoride. We used search terms like "fluoride enzyme activity", "toxicity", "neurotoxicity", "intelligence", "hormonal disruption", "oxidative stress", "fluorosis", etc. The searches were in some cases narrowed by limiting to "human".

We disregarded studies where no more than an abstract in English was available.

## 3. Mechanisms of Fluoride Toxicity on Cellular Level

Sodium fluoride has been used in laboratory studies of enzyme-controlled reactions since the beginning of the 20th century [3,13]. These studies contributed to the discovery of fundamental biochemical processes such as glycolysis, the citric acid cycle, lipolysis, and ion transport across membranes. Simultaneously, they demonstrated fluoride as an enzyme disruptor. In this review, we show common molecular mechanisms of potential fluoride toxicity. We assess the impact of the pluripotent toxic effects of fluoride on a multisystem level.

### 3.1. Fluoride Inhibits Enzymes through Competition with Magnesium

Fluoride in 0.01% concentration has an appreciable effect in inhibiting the anaerobic production of energy from glucose. Of the salts examined only fluoride inhibits glycolysis in low concentrations in various tissues [14]. Otto Warburg and Walter Christian have shown that this is due to the inhibition of enolase [15].

Enolase is a dimeric metal-activated enzyme, which uses two magnesium ions ($Mg^{2+}$) per subunit [16]. The inhibitory effect of fluoride on enolase through competition with $Mg^{2+}$ has been demonstrated in various cells and tissues such as hepatocytes and liver, muscle, red blood cells, kidney, intestine, testis and embryonic cells, brain, retina, and others [3,16,17]. The extent of fluoride inhibition depends on the concentration of $Mg^{2+}$ and phosphate.

The mechanism of fluoride inhibition of enzymes through competition with $Mg^{2+}$ has been accepted as the common mechanism of fluoride toxicity since $Mg^{2+}$ is the activator of more than 300 enzymes in humans [18].

Magnesium does not form stable compounds with fluorides in water at sub-molar concentration. Nevertheless, Antonny et al. found trifluoromagnesate ($MgF_3^-$) in kinases, mutases, phosphatases, and hydrolases with catalytic $Mg^{2+}$. These enzymes catalyze the transfer of phosphate [19–21]. Trifluoromagnesate is a new transition state analogue for phosphoryl transfer.

### 3.2. Fluoride Inhibits Phosphoryl Transfer Reactions

Several vital reactions, such as the syntheses of nucleic acids, proteins, lipids, and polysaccharides are driven by adenosine triphosphate (ATP) hydrolysis to adenosine monophosphate (AMP) and pyrophosphate (PPi). In adult men, the production of PPi via these reactions reaches several kilograms per day. Most tissues contain highly active pyrophosphatases (PPases), which catalyze the simplest phosphoryl transfer reaction, the hydrolysis of the symmetrical PPi substrate to two molecules of inorganic phosphate (Pi) (Figure 1). A fluoride ion is a potent and specific inhibitor of cytoplasmic PPases [22,23].

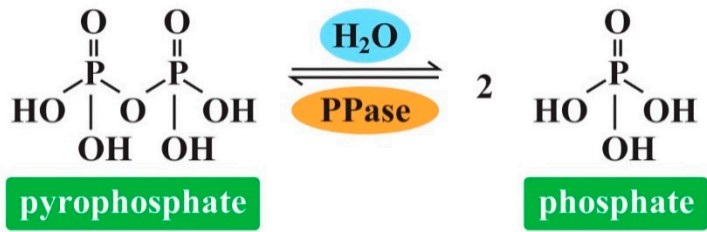

**Figure 1.** The hydrolysis of pyrophosphate to inorganic phosphate.

Intracellular PPi is a by-product of more than 200 different enzyme reactions and its hydrolysis plays a major role in driving fundamental biochemical reactions [23]. PPases provide a simple model for the studies on how the mechanism of fluoride affects the enzyme-catalyzed reversible phosphoryl transfer to the water.

### 3.3. Fluoride Inhibits Adenosine Triphosphatases by Multifactorial Mechanisms

The membrane integral PPases couple the hydrolysis of PPi to sodium ($Na^+$) and/or $H^+$ pumping. Fluoride inhibits the membrane-bound, proton-pumping $H^+$-ATPase, which is involved in the generation of proton gradients through the efflux of protons from the cell at the expense of ATP. Thus, fluoride has the dual action of dissipating proton gradients and preventing their generation through its action on $H^+$-ATPase. The collapse of the transmembrane proton gradient, in turn, reduces the ability of cells to transport solutes via mechanisms involving proton driving force [23].

P-type adenosine triphosphatases (ATPase) share common phosphorylation and dephosphorylation mechanism. The reactions of ATP binding, phosphoryl transfer and hydrolysis, and the transduction of the energy to the ion-binding site, are highly conserved. The transfer of a phosphoryl group regulates the activity of enzymes in the cell [24].

$Na^+$, $K^+$-ATPase (NKA) is an integral protein in the plasma membrane that transports $Na^+$ ions to the outside and $K^+$ ions to the inside of the cell [25]. By using the energy from ATP to establish asymmetric distributions of ions across the cell membrane, NKA links metabolic energy to cellular functions and to signaling events both between and within cells. More than 60 papers describe the effects of fluoride on the NKA activity, for example, in erythrocytes, hepatocytes and the liver, neurons and the brain, kidney, cardiac muscle and the heart, and adipocytes in both laboratory animals as well as humans. Declan T. Waugh explains the molecular mechanisms by which fluoride inhibits NKA [25]. He suggests that fluoride can inhibit NKA activity by lowering serum $Mg^{2+}$. Animal studies have also shown that fluoride exposure results in decreased serum $Mg^{2+}$ due to the decreased $Mg^{2+}$ absorption in the gastrointestinal tract. NKA activity is also downregulated by the activity of adenylyl cyclase, various hormones, cytokines, and neuropeptides [25]. NKA is a member of the P-type family of active

cation transport proteins, such as the $Ca^{2+}$-ATPase or the gastric $H^+$, $K^+$-ATPase [26]. Mechanisms of fluoride inhibition of ATPases activities are, therefore, multifactorial.

### 3.4. What Do the Studies of Fluoride Effects on Enzymes In Vitro Tell Us?

Biochemists clearly demonstrated that fluoride is an enzyme disruptor. Tables 1 and 2 show that fluoride both inhibits and activates a wide range of isolated enzymes from various tissues in vitro. The introduction of CWF evoked a question of the safety of the consumption of fluoride, which is an enzyme poison.

In 2006, a National Research Council (NRC) report stated that the available data were not sufficient to draw any conclusions about potential effects or risks to liver enzymes from low-level long-term fluoride exposures such as those seen with CWF. The concentrations of fluoride used in laboratory studies are several times greater than the concentration present in the body [27]. In agreement with this report, the American Dental Association (ADA) concluded that the recommended levels of fluoride in drinking water do not affect human enzyme activity in a living organism [28].

Such arguments have recently appeared in the comprehensive review of Guth et al. [29]. These authors stated that most of the effects of fluoride remain to be established and many of the findings from in vitro studies were only observed in the millimolar range. The in vivo relevance of such concentrations in humans is questionable since fluoride plasma concentrations in healthy adults generally range between 0.4 and 3.0 μM [29]. The findings of other studies found that plasma fluoride concentrations resulting from long-term ingestion of 1–10 ppm fluoride in drinking water range between 1–10 μM. Fluoride plasma concentrations in persons in endemic areas may reach levels of 7.37–39.5 μM [30].

The review of laboratory studies on the effects of fluoride on enzyme activities reveals that competing reactions might produce anomalous "paradoxical" effects. This means that the inhibitory or stimulatory impact of fluoride can differ with its concentration, i.e., inhibitory impact could be greater at a lower than at a higher concentration [31]. For example, Zakrzewska et al. [32] found that the activity of lactate dehydrogenase in ram semen displayed a nine-fold decrease with 20–200 μM fluoride, but at the concentration of millimolar fluoride, its activity is nearly 40% above that of the control. A stimulatory or beneficial effect of a subinhibitory concentration of a toxic substance is known as hormesis [33]. The existence of the paradoxical dose-responses further contributes to the various pattern of fluoride effects on enzymes both in vitro and in vivo.

**Table 1.** Inhibitory effect of fluorides on enzymes in vitro.

| ENZYME | SOURCE | FLUORIDE Concentration | REFERENCES |
|---|---|---|---|
| acid phosphatase | ram semen<br>osteoblasts<br>osteoclasts<br>kidney | 20–200 μM<br>mM<br>mM<br>10 μM | [32]<br>[34]<br>[35,36]<br>[37] |
| aconitase | liver | mM | [38] |
| adenylyl cyclase | liver<br>fibroblasts | 5–10 mM<br>5 mM | [39]<br>[40] |
| acetylcholinesterase | red blood cells<br>brain | 0.01–10 mM<br>5–50 mM | [41]<br>[42] |
| amylase | human saliva | 50–500 mM | [43] |
| arginase | liver, kidney | >4 mM | [44] |
| Ca-ATPase | sarcoplasmic reticulum | 5–10 mM | [26] |
| catalase | red blood cells<br>macrophages | μM<br>0.5–05 mM | [45]<br>[46] |

**Table 1.** *Cont.*

| ENZYME | SOURCE | FLUORIDE Concentration | REFERENCES |
|---|---|---|---|
| **cytochrome-c-oxidase** | liver, heart | 0.01–10 mM | [47] |
| **enolase** | red blood cells<br>hepatocytes<br>embryonic cells<br>oral bacteria | 1–50 mM<br>3 mM<br>50 µM<br>16–54 µM | [48,49]<br>[50]<br>[51]<br>[52] |
| **F(1)F(o)-ATP synthase** | mitochondria | mM | [53] |
| **glucose-6-phosphatase** | liver | µM | [19,54] |
| **glutathione peroxidase** | red blood cells | µM | [45] |
| **glycogen synthase** | hepatocytes | 2–15 mM | [55] |
| **lactate dehydrogenase** | ram semen<br>fetal osteoblast<br>bone marrow | 20–200 µM<br>6–60 µM<br><0.5 mM | [32]<br>[56]<br>[36] |
| **lipase** | pancreas, liver | 5 µM–5 mM | [57,58] |
| **L-Ca$^{2+}$ channels** | heart | 10 mM | [59] |
| **Na$^+$/K$^+$ ATPase** | plasma membrane | 1–10 mM | [25] |
| **protein phosphatase** | liver<br>bone | 10–50 mM<br>µM | [60]<br>[24] |
| **pyrophosphatase** | yeast | 5 mM | [24,61] |
| **pyruvate kinase** | red blood cells | 10–50 mM | [62] |
| **succinate dehydrogenase** | heart, liver, kidney | 1–15 mM | [63,64] |
| **superoxide dismutase** | red blood cells | µM | [45] |
| **urease** | human | 250 mM | [58,65] |

**Table 2.** Stimulatory effect of fluorides on enzymes in vitro.

| ENZYME | SOURCE | FLUORIDE Concentration | REFERENCES |
|---|---|---|---|
| **acid phosphatase** | ram semen | 100 mM | [32] |
| **adenylyl cyclase** | heart, liver, brain<br>smooth muscle<br>lymphoma cell<br>kidney | 1–10 mM<br>10 mM<br>10 mM<br>5 mM | [47,66–68]<br>[60]<br>[69]<br>[70] |
| **alkaline phosphatase** | bone cells | 10–100 µM | [71] |
| **aspartate transaminase** | ram semen | 20–200 µM | [32] |
| **Ca$^{2+}$-ATPase** | sarcoplasmic reticulum | 1–10 mM | [72] |
| **glutamyl S-transferase** | ram semen | 20–200 µM | [32] |
| **K$^+$[ACh]$_M$ channel** | heart | >1 mM | [73] |
| **K$^+$ATP channel** | heart | mM | [74] |
| **lactate dehydrogenase** | hepatocytes<br>ram semen | 1–30 mM<br>100 mM | [50]<br>[32] |
| **L-type Ca$^{2+}$ channel** | rabbit femoral artery | 10 mM | [60] |
| **glycogen phosphorylase** | hepatocytes | 1–50 mM | [39,75] |
| **tyrosine kinase** | osteoblasts | 1–10 mM<br>10–200 µM | [76]<br>[71,77] |

*3.5. Fluoride Effects on Transmembrane Signaling: A Link to Understanding Integrative Networks in Fluoride Toxicity*

Fluoride played an important role as a tool in the understanding of hormone signaling cascades in the second half of the last century. The liver membranes and multi-receptor fat cell systems provided the first insight that adenylyl cyclase (AC) is involved in the transduction of hormonal signals (Tables 1 and 2). Theodore Rall and Earl W. Sutherland discovered that AC synthesizes the cyclic AMP (cAMP) from ATP. This observation led to the discovery of G proteins and intracellular second messengers' cascades [66,78]. Alfred G. Gilman and Martin Rodbell were awarded The Nobel Prize in Physiology or Medicine 1994 for their discovery of G-proteins and their role in signal transduction. Rodbell suggested in his Nobel speech that the system may contain a Mg-ATP complex at the catalytic site and fluoride markedly reduced the concentration of $Mg^{2+}$ necessary for stimulation of AC activity (Figure 2) [79].

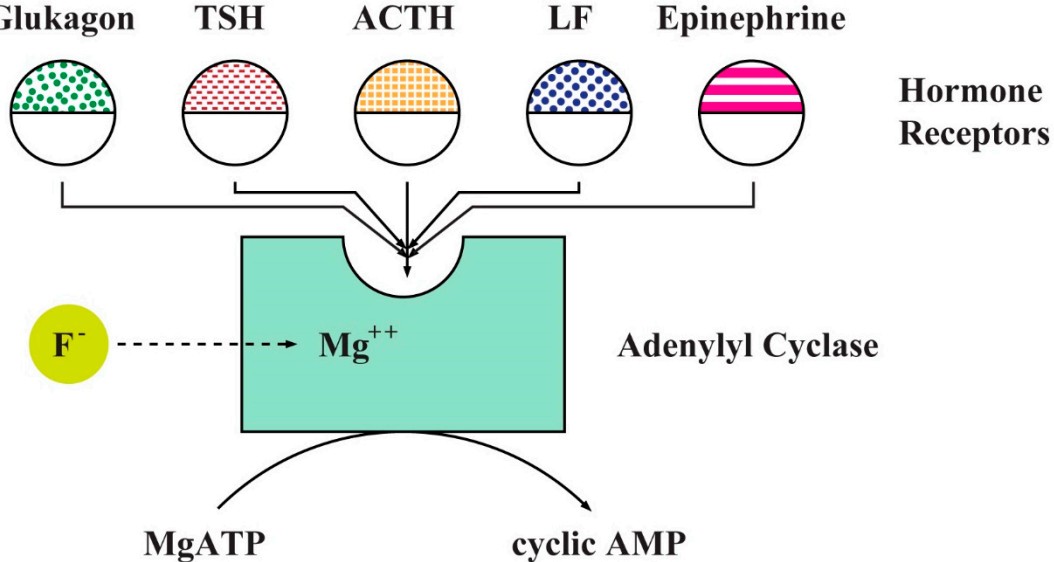

**Figure 2.** The effect of fluoride on the adenylyl cyclase according to Martin G. Rodbell (https://www.nobelprize.org/prizes/medicine/1994/rodbell/lecture/79).

3.5.1. G Protein Signaling Cascade

Heterotrimeric G protein is formed by alpha (Gα), beta, and gamma subunits (Gβγ) [80–82]. The name of G protein was selected because the Gα subunit binds guanosine diphosphate (GDP). A hormone-activated receptor triggers the exchange of GDP to bind guanosine triphosphate (GTP) in the Gα, the alpha subunit dissociates from the Gβγ and interacts with AC, which is an effector molecule. The result of a hormone signal binding to its receptor is elevated concentration of cAMP in the cell (see Figure 3).

All eukaryotes use G proteins for signaling and have evolved a large diversity of G proteins. For example, humans encode 18 different Gα proteins, 5 Gβ proteins, and 12 Gγ proteins. There are many classes of Gα subunits, such as Gαs (activation of AC), Gαi (inhibition of certain AC isoforms), and Gαq/11 (phospholipase C activation) [83,84]. The diversity of G proteins and effector molecules broaden enormously the possibilities of molecular interactions during signal transduction.

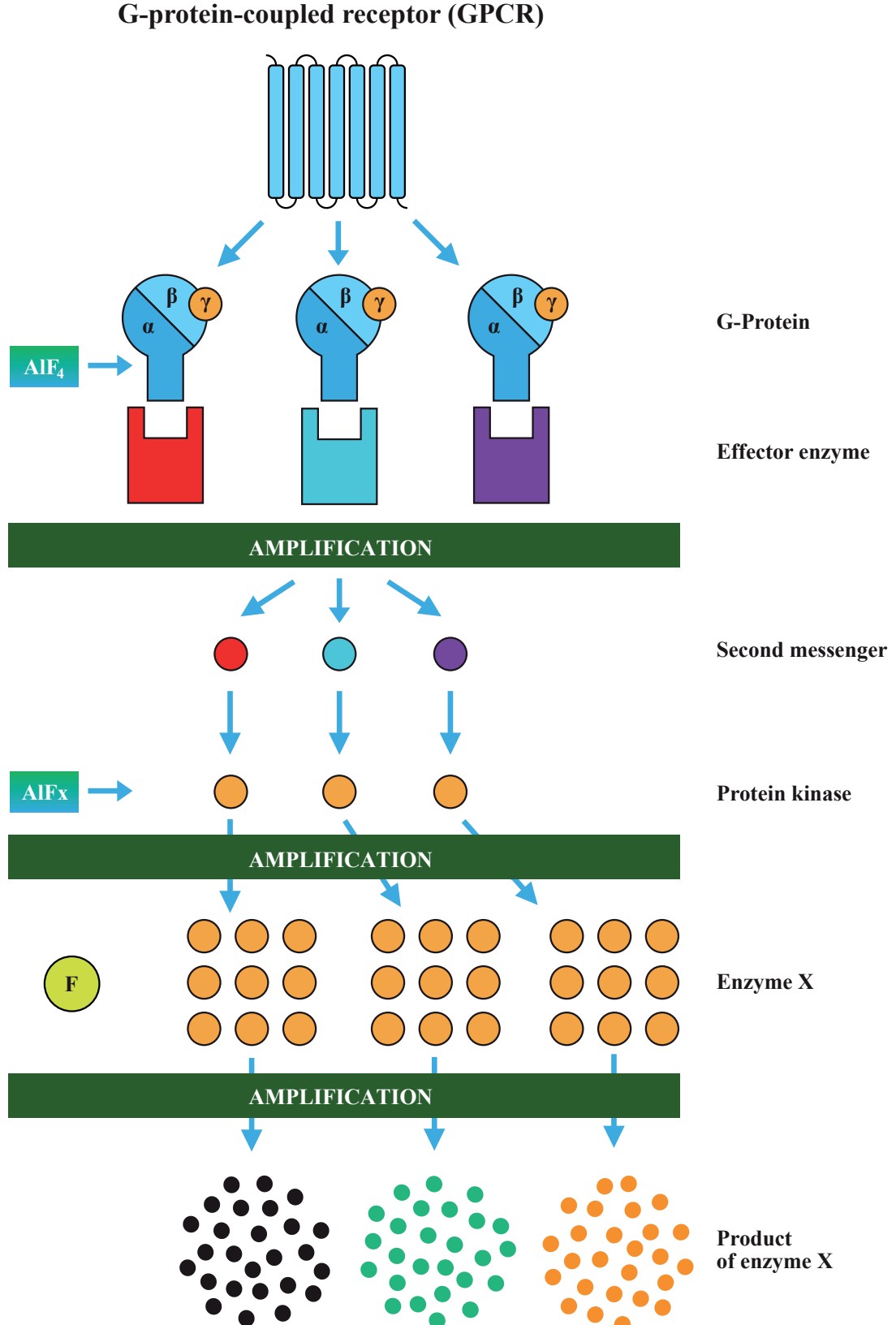

**Figure 3.** The message of AlFx is greatly amplified in G protein signaling cascade. Effector enzymes are adenylyl cyclase (AC) or phospholipase C, the second messenger molecule could be cAMP, inositol 1,4,5-trisphosphate, and diacylglycerol.

### 3.5.2. Aluminofluoride Complexes

In the 1980s, Gilman noticed that the activation of G protein by fluoride depends on trace amounts of free aluminum ions ($Al^{3+}$), a contaminant in glass test tubes, and chemicals used in AC assays [67]. The hypothesis that the activating species is aluminum tetrafluoride ($AlF_4^-$), which acted as an analog of a $\gamma$ phosphate moiety in $G\alpha$ GTP, was confirmed by $^{19}F$ NMR (nuclear magnetic resonance) titration experiments [67,80,85]. X-ray crystallography also provided evidence that $AlF_4^-$ seems to be the active site species [86]. Further modeling of the equilibria complexes predicted that at physiological pH, in millimolar fluoride concentrations, the ternary complex $AlF_3(OH)^-$ and neutral $AlF_3$ predominate [87–89]. These discoveries heralded a new field of research of aluminofluoride complexes (AlFx), which mimicked the phosphate group ($PO_4^{3-}$) in phosphate transfer reactions [89]. However, the exact structure of the activating species in solution is still being debated [20,82]. Using the $^{19}F$ NMR analysis and computational chemistry, Jin et al. recently showed that $AlF_4^-$ is the primary species, which mimics the transition state of the reaction [20].

The widespread use of AlFx as a general activator of heterotrimeric G proteins provide evidence that AlFx is a molecule giving false messages, which are amplified by processes of signal transduction (Figure 3).

A small false signal of AlFx can activate various $G\alpha$ and effector proteins, such as AC or phospholipase C. In consequence, this generates cAMP, inositol 1,4,5-trisphosphate, and diacylglycerol. The increased level of second messenger-molecules activates protein kinase A, which phosphorylates a great number of proteins. AlFx acts again as the analog of phosphate. Protein phosphorylation constitutes one of the major posttranslational mechanisms employed in the signaling cascades. Phosphoryl transfer reactions are also involved in processes such as regulation of cell metabolism, energy transduction, cytoskeletal protein assembly, regulation of cell differentiation and growth, aging, and apoptosis. Considering that all these reactions are fundamental for nearly all biological processes, the common denominator of which is the transfer of a phosphoryl group, we can conclude that fluoride represents the dangerous hidden toxin for all living organisms.

### 4. The Implication of Fluoride Toxicity on Human Health

Early reports about fluoride toxicity in humans appeared in the journal Physiological Reviews already in 1933 [58]. Aluminum smelter workers and persons living near the factory where fluoride was in high concentration in the atmosphere suffered with psychiatric and neurological disturbances. Endemic fluorosis is caused by persistent fluoride exposure through ingestion or inhalation, and most commonly, as a result of high fluoride levels in drinking water and beverages [90,91]. The WHO's drinking water quality Guideline Value for fluoride is 1.5 mg/L [92].

Dental fluorosis results after excess fluoride ingestion, most commonly in drinking water, during tooth formation. For example, dental fluorosis appears in 43–63% of schoolchildren in endemic areas of China with total fluoride intake 2.7–19.8 mg/day [5]. The rate of dental fluorosis also increases in countries with CWF. In the 2010–2012 survey, dental fluorosis was reported among adolescents aged 12–15 years in the USA with a surprisingly high rate of 65% [93]. Dental fluorosis might be used as a sensitive indicator of excessive fluoride exposure [94].

Skeletal fluorosis impacts millions of people in regions with high natural levels of fluoride, like India, China, Pakistan, Iran, and the Gulf region, to name a few [91]. The U.S. Environmental Protection Agency (USEPA) sets a maximum contaminant level of 4.0 mg/L to protect against skeletal fluorosis. WHO indicates a clear risk of skeletal fluorosis for a total intake of 14 mg fluoride per day. Nevertheless, the recent findings revealed that consumption of fluoride at even 10 times lower concentrations of 1.5 mg/L caused its high incidence in India [95]. The chemical analyses show that 80% of water sources in rural areas exceed the WHO fluoride permissible limits and residents are affected by skeletal fluorosis.

Waldbott et al. examined about 500 people affected by chronic fluoride intake from CWF [96]. These authors observed chronic fatigue, headaches, loss of the ability to concentrate, depression, gastrointestinal symptoms, and deterioration of muscular coordination.

### 4.1. Oxidative Stress

Under normal physiologic conditions, glutathione (GSH) regulates many functions such as reactive oxygen species (ROS) scavenging, maintaining cell membrane integrity, signal transduction, etc. Intracellular oxidative stress may have functional consequences such as increased mitochondrial superoxide production and a chronic inflammatory response. Excessive intracellular oxidative stress may contribute to several body dysfunctions like diabetes, neurological diseases, cancer, and rapid aging. The imbalance between oxidants and antioxidants is regulated by the GSH redox system in the human body. Some authors suggest that oxidative stress is the key mechanism of fluoride toxicity [95,97–99]. Fluoride causes the increased generation of ROS such as superoxide anion, reactive hydroperoxides (ROOH), and hydrogen peroxide ($H_2O_2$) (Figure 4). Fluoride exposure can reduce the cellular level of GSH, often resulting in excessive production of ROS [97–99]. Fluoride inhibits the activity of antioxidant enzymes such as catalase, glutathione peroxidase (GPx), GSH reductase (GR), and superoxide dismutase (SOD) (see Table 1).

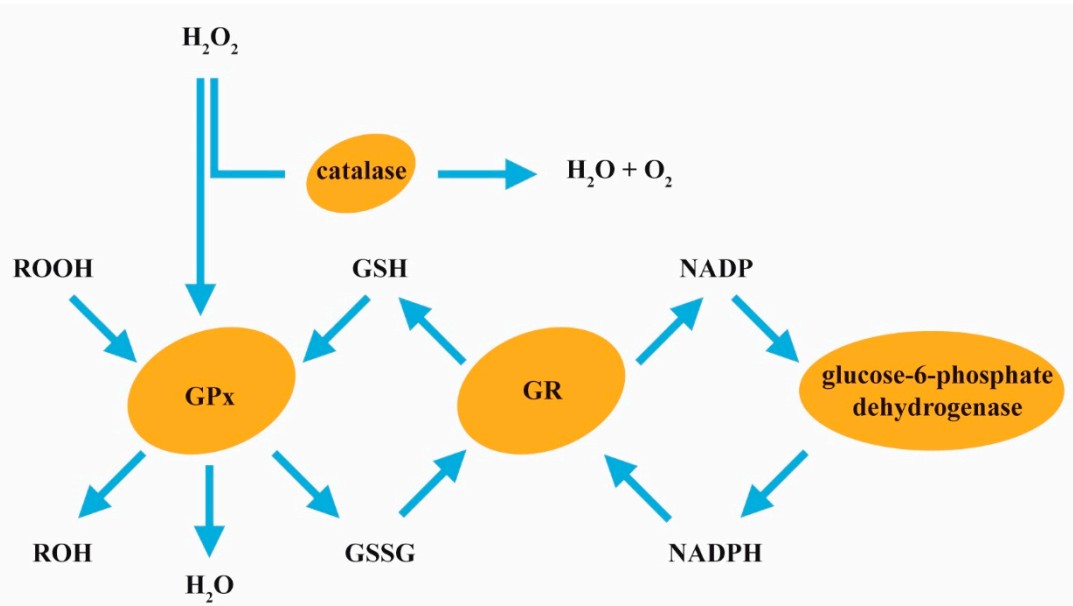

**Figure 4.** The glutathione (GSH) redox system. Hydrogen peroxide ($H_2O_2$), the immediate precursor of hydroxyl radical, is metabolized via the action of catalase and glutathione peroxidase (GPx). GPx also metabolizes reactive hydroperoxides (ROOH) and oxidizes reduced GSH to its disulfide form (GSSH), which is recycled back to GSH by the action of glutathione reductase (GR). A cofactor for GR is the reduced nicotinamide adenine dinucleotide phosphate (NADPH, which is supplied by the action of glucose-6-phosphate dehydrogenase).

It has been found that a high amount of fluoride can trigger oxidative stress by sabotaging the intracellular anti-oxidative defense system. The levels of ROS were significantly increased, while the activity of SOD was decreased in the blood of subjects from endemic fluorosis areas in China [5].

A close association between chronic fluoride toxicity and increased oxidative stress has been reported in children with skeletal fluorosis [95,100]. A significantly lower redox ratio of GSH to GSSG, as the marker of increased oxidative stress, has been reported in children with autism spectrum disorder (ASD). Increased vulnerability to oxidative stress may contribute to the development and clinical manifestation of ASD [101,102].

The increased activity of the enzyme glutathione S-transferase (GST) was found in a group of schoolchildren with the higher levels of dental fluorosis drinking water with 1.8 ppm fluoride in Mexico [103]. The authors suggested that the increased GST activity most likely was the result of the body's need to inactivate free radicals produced by exposure to fluoride.

### 4.2. The Implication of Fluoride Interactions with G Protein-Coupled Receptors for Human Health

The number of known G protein-coupled receptors (GPCRs) exceeds 800, which makes them the largest family of membrane proteins encoded in the human genome [83,84,104]. Malfunction of GPCRs functions is prevalent in human diseases. Albert Gilman said in his Nobel Prize lecture [80]: "The ultimate dream is to design drugs that will prevent aberrant G protein action." Recently, approximately half (estimates vary between 30% and 60%) of drugs marketed to improve health target GPCRs [105–107].

By contrast, fluoride in concentrations of $10^{-1}$–$10^{-6}$ M, in the presence of trace amounts of $Al^{3+}$, can affect all G proteins. AlFx may evoke several signaling disorders and act as the hormonal disruptor. AlFx affects levels of second messengers, calcium homeostasis, protein phosphorylation, cytoskeletal proteins, ion transport, the metabolism and functions of all blood elements, endothelial cells, blood circulation, the function of the immune system, bone cells, fibroblasts and lung cells, metabolism of the liver, processes of neurotransmission, functions of the brain including cognition, and mental acuity.

The severity and the development of symptoms depend on age, nutrition status, kidney function, and many other factors. Simultaneously, the heterogeneity of mutual dynamic interactions in signaling cascades can explain the clinically heterogeneous symptoms and contribute to an understanding that chronic exposure to fluoride might result in various symptoms of toxicity.

The synergy of AlFx with submaximal hormonal alterations results in an effective response having significant pathophysiological implications. Under such circumstances, a "safe" fluoride concentration might induce pathological effects in a person with genetic susceptibility.

### 4.3. Disturbances of the Thyroid Gland

Investigations from endemic areas demonstrate that the thyroid gland is sensitive concerning fluoride burden [108,109]. Thyroid-stimulating hormone (TSH) is a pituitary hormone that stimulates the thyroid gland to produce thyroid hormones thyroxin (T4) and triiodothyronine (T3). A high level of TSH indicates an underactive thyroid gland or hypothyroidism. The receptor for TSH belongs to the category of GPCRs. AlFx is able to mimic TSH by switching on its associated G protein. It is suggested that the consequent overproduction of cAMP leads to a feedback mechanism resulting in a desensitization of the TSH receptor and ultimately to a reduced activity of the thyroid gland.

The deficient thyroid function with elevated TSH at high fluoride exposure was reported in both children and adults in fluoride endemic areas in India since 2005 [110–113]. According to Susheela et al., 47% of children living in a New Delhi neighborhood with an average water fluoride level of 4.37 mg F/L have evidence of clinical hypothyroidism attributable to fluoride [111].

Wang et al. recruited 571 resident children aged 7–13 years from endemic and non-endemic fluorosis areas in Tianjin, China. They found that every 1 mg/L increment of water fluoride was associated with a 0.13 µIU/mL increase in TSH. [114]. Ruiz-Payan et al. found that teenagers drinking water with one ppm of fluoride in Northern Mexico have reduced T3 levels [115].

Every 1 mg/L increment of urinary fluoride was associated with 0.09 µg/dL decrease in T4 [115].

Kheradpisheh et al. [108] compared the average amount of T3, T4, and TSH in people with hypothyroidism and people without thyroid disease, concerning fluoride concentrations 0–0.29 and 0.3–0.5 mg/L in drinking water in Iran. The major conclusion of this study was that fluoride impacts TSH and T3 even in concentrations lesser than 0.5 mg/L.

A recent study from an endemic fluorosis village, Talab Sarai in Pakistan, with a high content of fluoride in the drinking water (6.23 mg F/L) found teeth fluorosis in 93% of 130 investigated children

with the age of 12 ± 3.3 years. Eighty percent of the children displayed clear thyroid hormonal derangements, with 37% having high TSH and 43% with T3 and T4 disorders [109].

The diagnosis of hypothyroidism was nearly twice as frequent in fully fluoridated areas in England, as compared to non-fluoridated areas [116]. The population-based study of the weighted sample of 6,914,124 adults aged 18–79 living in Canada who have higher levels of urinary fluoride also shows increased risk for underactive thyroid gland activity [117].

Such epidemiological studies bring strong evidence that thyroid dysfunction appears in both endemic fluorosis areas and in areas with CWF [94,118].

### 4.4. Melatonin

Melatonin, the hormone produced by the pineal gland at night, regulates large numbers of life processes, such as the sleep-wake cycle, reproduction, development, and aging. It is also a modulator of mitochondrial metabolism, digestive functions, and immunity (Figure 5). Because of its radical scavenging actions melatonin reportedly restrains cancer development and growth [119].

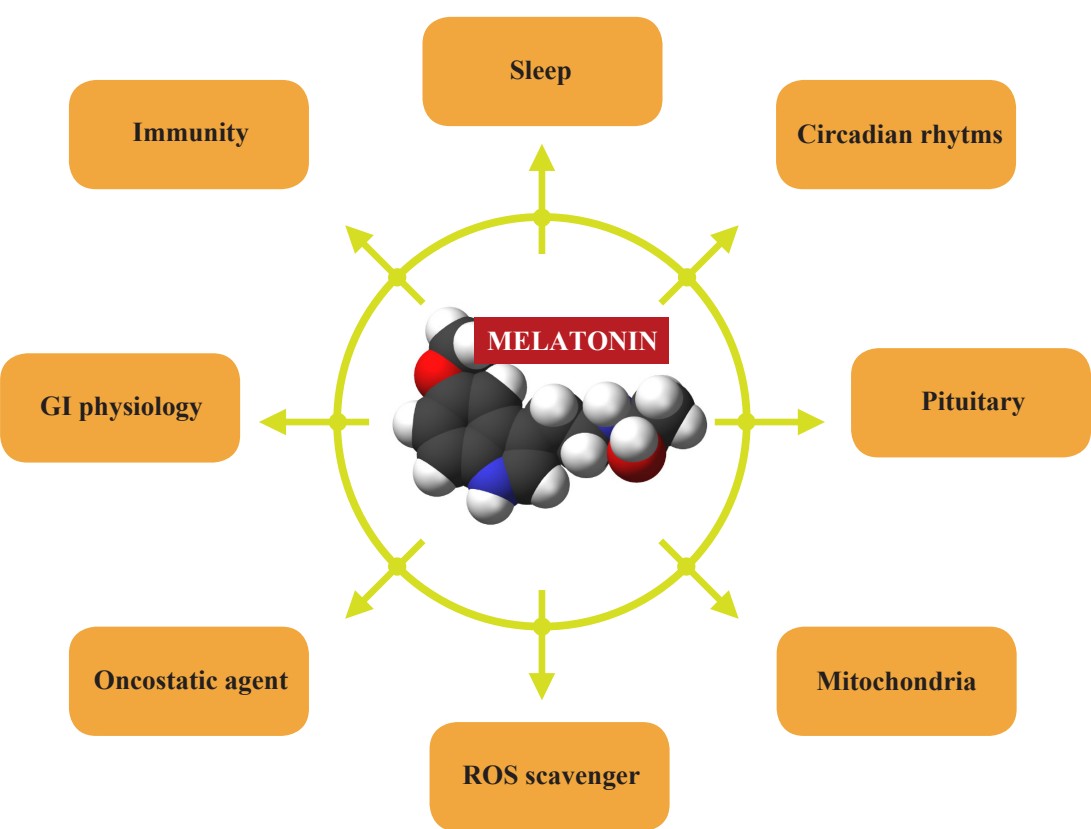

**Figure 5.** Physiological functions of melatonin and fluoride interventions. File melatonin has been used from the Wikimedia Commons according to the GNU Free Documentation License.

Luke studied the effect of fluoride from drinking water on gerbils during sexual maturation [120]. Animals excreted less melatonin metabolite in their urine and took a shorter time to reach puberty. Luke also analyzed the pineal glands from 11 human corpses and found that fluoride in the apatite crystals averaged about 9000 ppm [121]. Based on this evidence fluoride is likely to cause decreased melatonin production and to have other effects on normal pineal function, which in turn could contribute to a variety of effects in humans [122].

The Newburgh–Kingston caries-fluoride study compared the development of girls aged 7–18 years in fluoridated Newburgh (1.2 mg F/L) with girls in fluoride-free Kingston. The investigators found

that girls in Newburgh had menarche earlier than girls in Kingston [123]. The authors suggested an analogy with Luke's observations in gerbils [120].

Melatonin has been recognized by the European Medicines Agency (EMA) since 2007 for primary insomnia in adults over 55 years of age. Pooled data demonstrate that exogenous melatonin lowers sleep onset latency and increases total sleep time, whereas it has little if any effect on sleep efficiency [124]. Children and young people with significant sleep issues are often prescribed melatonin, namely the subjects with ASD [102]. Insomnia and sleep disorders occur in 50–80% of ASD patients [125,126]. The disruption of the serotonin-N-acetyl serotonin-melatonin pathway has been found in a group of 278 patients with ASD [127–129]. Pagan et al. suggested the levels of plasma melatonin, whole-blood serotonin, and platelet N-acetyl serotonin as biomarkers for ASD. They found a deficit in melatonin in 51% as well as hyperserotonemia in 40% of ASD patients. [127,128]. Serum serotonin was also increased in a cohort of healthy children drinking water containing 2.5 ppm fluoride, as compared to controls [102,129].

The study of fluoride exposure and sleep patterns among a sample of adolescents living in the USA shows that fluoride exposure may be a risk factor for sleep disturbances [130]. The authors found that each 0.52 mg/L increase in tap water fluoride concentration was associated with 1.97 times higher odds of participants reporting snorting, gasping, or stopping breathing while sleeping at night.

Two melatonin receptors have been identified in humans. They are both GPCRs and can be affected by AlFx. Regardless of mechanism of the effect of fluoride on melatonin, whether this is primary via melatonin receptors or secondary through pineal calcification, its intervention could have the wide spectrum of pathological effects [119]. Additional prospective human studies are needed to explore the impact of fluoride on melatonin for healthy physical and mental development and aging [131,132].

### 4.5. Fluoride-Induced Neurotoxicity

Interest in the developmental neurotoxicity of fluoride has grown significantly since the 2016 report of the National Toxicology Program (NTP) on Fluoride Toxicity that recommended the USEPA set a new drinking water standard—approximately 0.03–1.5 ppm. Recently, the NTP released a monograph on fluoride concluding that fluoride is presumed to be a cognitive neurodevelopmental hazard [133].

While there are thousands of articles pointing to the safety of CWF, there is robust evidence on fluoride neurotoxicity. Using the PubMed database and the web site of the journal Fluoride, we found 315 laboratory, clinical, and epidemiological studies over the whole world providing evidence about fluoride neurotoxicity. This is in good agreement with Hirzy et al., who reported over 300 animal and human studies considering fluoride neurotoxicity [4]. The comprehensive historical review of fluoride neurotoxicity in humans provided Spittle [134,135]. There is a compelling evidence that children living in endemic fluorosis areas have a statistically lower IQ than those who live in a low fluoride level area [94,136].

### 4.5.1. Mechanisms of Fluoride Neurotoxicity on the Cellular Level

Fluoride has been shown to inhibit glycolytic enzymes in vitro, with a loss of mitochondrial membrane potential, cytochrome c release, and apoptosis [137]. When neuronal cellular energy production and $Mg^{2+}$ are deficient, excitotoxicity is greatly enhanced—so much so that even physiological levels of glutamate or other excitatory amino acids can produce excitotoxicity [138]. Excitotoxicity plays a major role in most neurodegenerative diseases [102]. Likewise, both fluoride and AlFx are known to increase brain oxidative stress. Rat hippocampal neurons cultured in the presence of fluoride showed a significant decrease in GPx activity and decreased GSH concentration. Because of the intimate link between elevations in brain ROS and LPP products and excitotoxicity, we can be confident that the latter process most likely plays a critical role in fluoride neurotoxicity [139]. In the brain, fluoride affects cellular energy metabolism, synthesis of inflammatory factors, neurotransmitter

metabolism, microglial activation, and the expression of proteins involved in neuronal maturation. With knowledge of the pluripotent mechanism of fluoride toxicity on the molecular and cellular level, we can also predict its adverse effects on the brain during aging [140,141].

### 4.5.2. Developmental Neurotoxicity of Fluoride in Children

Since the evidence of fluoride in the cerebrospinal fluid of patients with fluorosis in 1988 [142], imaging studies of radioactive fluoride showed that fluoride passes the blood-brain barrier (BBB) [94]. Based on the research from China, the fetal brain is highly susceptible to fluoride poisoning since the fetal BBB is immature and readily permeable to fluoride.

A study by Du et al., performed in 1992, revealed the effects of fluoride on the brains of 15 fetuses from an endemic fluorosis area compared with those from a non-endemic area [143]. These studies showed delayed brain development in endemic fluorosis areas. Du et al. concluded that the passage of fluoride through the placenta of mothers impacts brain development.

The effect of fluoride chronic exposure on children's intelligence quotient (IQ) has been used as an indication of the neurotoxic effect of fluoride. Several studies published in China, Iran, India, and Mexico found an association between lowered IQ and exposure to fluoride [136,144–150].

Most of the robust evidence has been gathered in China. In many communities the daily intake of fluoride exceeds two milligrams [151]. A meta-analysis of 16 studies carried out in China between 1998 and 2008 found that children living in an endemic area have five times higher odds of developing a statistically lower IQ than those who live in a low fluoride level area [152]. In 2012, Choi et al. [153] assessed a meta-analysis of 27 research reports published over 22 years. The last prospective birth cohort studies from Mexico and Canada provided evidence that intrauterine exposure to fluoride is most important in developmental fluoride neurotoxicity. The effects of prenatal fluoride exposure and development of children´s cognitive abilities were followed in 299 Mexican mother–children pairs of the Early Life Exposures to Environmental Toxicants birth cohort study [150]. Higher levels of fluoride in mothers' urine during pregnancy were associated with lower cognitive and IQ scores in their children. The authors estimated that each 0.5 mg F/L increase in maternal urinary concentration was associated with an average decrease of 3.15 and 2.50 points in cognitive and IQ scores, respectively. A further study on 213 mother–child pairs of the same project in Mexico City found that higher prenatal fluoride exposures during pregnancy were associated with global measures of Attention Deficit Hyperactivity Disorder (ADHD) and more symptoms of inattention in the offspring [154].

In the Canadian study, maternal exposure to higher levels of fluoride during pregnancy was associated with lower IQ scores in children at 3–4 years [155]. An increase of fluoride in maternal urine for 1 mg/L was associated with a 4.5 point decrease of IQ score in boys.

A recent 2020 study by Till et al. of Canadian cohort children compared IQ scores in 398 children who were formula-fed versus breastfed during infancy [156]. This detailed analysis indicates that fluoride intake among infants younger than six months may exceed the tolerable upper limits if they are fed exclusively with formula reconstituted with fluoridated tap water. After adjusting for fetal exposure, the investigators found that fluoride exposure during infancy predicts diminished non-verbal intelligence in children.

Hirzy et al. [4] used data from a study by Xiang et al. of more than 500 children in China [157] and estimated a safe daily dose of fluoride. Based on their calculations, a protective daily dose should be no higher than 0.05 mgF/day, or 0.0010 mgF/kg-day for children.

The fluoride-induced lowered IQ quotient among children remains the most widely discussed adverse effect of fluoride by the scientific community [29,94,158]. Philippe Grandjean concluded that recent epidemiological results support the notion that elevated fluoride intake during early development can result in IQ deficits that may be considerable [94]. On the other hand, Guth et al. [29] found in their comprehensive review that, so far, the 21 of 23 studies investigating the effect of fluoride on intelligence are of low quality with some limitations leading to confounding effects related to a constellation of factors, including, in comparison to the "reference populations", the "exposed populations" being

in relatively poor rural communities with less-developed healthcare systems, lower educational and socioeconomic status, lower overall nutritional status and intake of essential nutrients, and higher exposure to environmental contaminants, such as lead, cadmium, mercury, and manganese.

A detailed assessment of 73 human studies based on IQ examinations is provided by Connett et al. [159]. These authors found that as of May 2020, a total of 73 studies have investigated the relationship between fluoride and human intelligence. Of these investigations, 63 studies have found the association of elevated fluoride exposure with reduced IQ in 23,872 children [94].

These findings support the requirement to reduce fluoride intake during pregnancy. Due to the limited opportunities for repair and compensation, any damage that occurs to a brain of a fetus or child will likely remain for the rest of his/her life [160]. However, most children affected will not receive a neurodevelopmental diagnosis, and the global occurrence of adverse effects has, therefore, recently been termed a "silent pandemic" [94].

### 4.5.3. The Role of Fluoride in the Etiopathogenesis of Autism Spectrum Disorder (ASD)

The high ASD prevalence in countries with CWF as well as in countries with endemic fluorosis supports our view that fluoride is an important environmental factor in ASD etiopathogenesis. We evaluated the role of chronic fluoride exposure regarding mitochondrial dysfunction, oxidative stress, immunoexcitotoxicity, and decreased melatonin levels [5,102,139]. These symptoms appear both after chronic fluoride exposure as well as in ASD. By contrast, some European countries, which rejected CWF shortly after its introduction in the 1970s–1990s, such as Germany, France, Poland, and the Czech Republic, report a low prevalence of ASD. In countries with CWF, such as Ireland and some areas in the UK, the prevalence of ASD is comparable with the USA, Canada, and China.

### 4.5.4. Fluoride in Fetal and Infant Development

It seems evident that fluoride offers no benefits to the fetus. The beneficial effects of fluoride predominantly occur at the tooth surface, after the teeth have erupted. The CDC concluded that fluoride supplementation during pregnancy did not benefit the child's dental health [11,161]. Accordingly, the Canadian Pediatric Society advises against fluoride supplements during this period [162]. The EFSA Panel did not estimate the Adequate Intakes for fluoride for the first six months of life [11].

We suggest the reduction of fluoride exposure of pregnant women and infants as an efficient way for prevention an IQ loss and ASD epidemic soon. Monitoring of the ASD prevalence in children born after the removal of fluoride from drinking water could provide relevant information for our hypothesis.

### 4.6. The Potential Risk of Novel Fluorinated Drugs

The advancements in organofluorine chemistry have increased the potential for the synthesis of fluorinated compounds used as novel drugs. These include general anesthetics, antibiotics, antiviral and antimalarial agents, anti-inflammatory drugs, antidepressants, antipsychotics, and different bio-compatible materials. Twenty percent of modern pharmaceuticals contain fluorine [163].

Fluorine substitution in a drug molecule influences not only its pharmacokinetic properties, such as absorption, tissue distribution, secretion, route and rate of biotransformation, but also its pharmacodynamics and toxicology. The fluorinated drugs are more potent; however, much less is known about the potential danger of these compounds for human health. We present some examples of fluorinated drug biotransformations, which could potentially contribute to the increasing burden of fluoride and evoke long-term health problems.

Metabolic defluorination can readily occur, which favors the formation of stable fluoride ion, despite the strength of the C–F bond [163]. The highest defluorinating activity is characteristic of the liver, with the kidney, lungs, heart, and testicles in order of decreasing activity.

The first of the modern fluorinated anesthetics halothane is metabolized by cytochrome P450 (CYP). In vitro investigations have identified a role for human CYPs 2E1 and 2A6 in oxidation and CYPs 2A6 and 3A4 in reduction [164]. The reductive pathway generates free radicals, which undergo

further reduction and fluoride elimination. Oxidation produces trifluoroacetyl (TFA) chloride capable of covalently binding to proteins. TFA can either form trifluoroacetic acid, and be excreted with fluorine in urine, or form TFA adducts, which act as antigens in halothane-induced hepatitis [165].

Another well-known example of a fluorinated drug is cerivastatin, from the statin class. The awareness of muscular adverse drug reactions and fatal kidney failure led to the withdrawal of cerivastatin from the market in 2001 because of 52 deaths attributed to drug-related rhabdomyolysis [166]. Rhabdomyolysis is a syndrome caused by injury to skeletal muscle, which involves leakage of large quantities of potentially toxic intracellular contents into the blood plasma. This can lead to serious renal failure. Rhabdomyolysis was 10 times more common with cerivastatin than the other five approved statins. Laboratory and clinical investigations show that cerivastatin exerts direct pleiotropic effects on vascular endothelium and muscle fibers. It caused a significant membrane hyperpolarization, increased cGMP levels, and nitric oxide production in cultured human umbilical vein endothelial cells [167]. These data show that cerivastatin activates endothelial $Ca^{2+}$-activated $K^+$ channels, which play an important role in the signaling of cerivastatin-mediated endothelial NO production and proliferation.

Fluoroacetate is a rare naturally occurring organofluorine that is surprisingly poisonous. Biochemists found that the toxicity was due to the CH2F group [168]. In addition to mice, rats, guinea-pigs, and rabbits, the larger animals, cats, dogs, monkeys, goats, and horses were used for studies of its toxicity. Fluoroacetate was studied for potential use as a chemical weapon during World War II. Fluoroacetate is a simple molecule capable of inactivating the citric acid cycle with the substitution of a single hydrogen substituent for fluorine in the CH2F group (Figure 6).

**Figure 6.** Fluoroacetate combines with coenzyme A to form fluoroacetyl CoA, which reacts with citrate synthase to produce fluorocitrate.

Despite understanding of the biochemical mechanism of fluoroacetate toxicity, an efficient antidote preventing the fluorocitrate synthesis has not been found. Surprisingly, ethanol, if taken immediately after the poisoning, has been the most efficient antidote [168]. Some fluorinated compounds—used as anticancer agents, narcotic analgesics, pesticides or industrial chemicals—metabolize to fluoroacetate as intermediate products. It was detected in the urine of oncologic patients with solid cancers treated with 5-fluorouracil (5-FU) [169,170]. Its metabolites accumulate inside cells and inhibit tumor growth. A pharmacologically active 5-fluoro-deoxyuridine monophosphate inhibits thymidylate synthetase and thus the formation of thymidine. The metabolization of 5-FU by scission of the pyrimidine ring leads to the formation of the highly toxic fluoroacetate. Polk et al. [171] identified 26 studies concerning 5-FU-induced cardiotoxicity, which is the main adverse effect on patients. These authors admit that it is likely that fluoroacetate plays a key role in 5-FU-induced cardiotoxicity.

## 5. The Economic Consequences of Fluoride Toxicity

It is difficult to precisely evaluate the economic consequences of endemic fluorosis. It is now known to be global in scope, occurring on all continents. Except for China, defluorination projects were closed in most countries due to a collapse in their cost.

Conversely, CWF is recognized as one of the most cost-effective, equitable, and safe measures to prevent cavities and improve oral health. Thousands of studies showed that CWF prevents cavities and

saves money, both for families and the health care system. An economic review of multiple studies used by CDC found that savings for communities ranged from $1.10 to $135 for every $1 invested. Per capita annual costs of CWF range from $0.11 to $24.38, while per capita annual benefits range from $5.49 to $93.19 [172]. Fluoride has modest benefit in terms of reduction of dental caries but also significant costs concerning dental and skeletal fluorosis, hypothyroidism, and mental and cognitive disturbances. Ingestion of fluoride constitutes an unacceptable risk with virtually no proven benefit. The possibility that chronic fluoride intake might evoke chronic diseases with high socioeconomic impact must be involved. Hirzy et al. calculated that the economic impact of IQ loss among USA children is the loss of tens of billions of dollars [4]. The available information supports a reasonable conclusion that economic losses associated with ASD may be also quite large. The annual societal costs for children with ASD were estimated between $11.5–60.9 billion in the USA in 2011. They included a variety of direct and indirect costs, from medical care to special education, and lost parental productivity [173]. Children and adolescents with ASD had average medical expenditures that exceeded those without ASD by $4110–6200 per year. According to estimates of Leigh and Du, annual costs due to ASD in the USA in 2015 were around $268 billion [173]. However, if the rate of increase in the ASD prevalence continues, Cakir et al. estimated in 2020, that costs could reach nearly $15 trillion by 2029 [174].

We suggest that the reduction of fluoride from the daily life of pregnant women as well as of children in infancy could be an efficient way to prevent fluoride developmental neurotoxicity [138].

## 6. Conclusions

Fluoride toxicity has been demonstrated in many studies. We present evidence that fluoride interferes with enzyme activities, induces oxidative stress, and causes hormonal disturbances, and neurotoxicity. The health impacts of increased fluoride exposition of millions of people in endemic fluorosis areas are of global public concern. At present, there is a divergence between the practice of CWF, which is regarded as valuable and safe for reducing dental caries, and current scientific evidence, which indicates that fluoride is a potent neurotoxin disturbing prenatal as well as postnatal brain development. The dose recommended for dental caries reduction is close to the dose causing pathological effects. Moreover, fluoride in synergy with trace amounts of $Al^{3+}$ has the potential to affect all phosphoryl transfer reactions and most regulations of fundamental biological processes in much lower concentrations than fluoride alone.

The potential neurotoxicity associated with exposure to fluoride has generated controversy about CWF. Given the large number of studies showing cognitive deficits associated with elevated fluoride exposure under different settings, the general tendency of fluoride-associated neurotoxicity seems overwhelming. We concur with the conclusions of many authors over the world that fluoride neurotoxicity is a serious risk associated with elevated fluoride exposure, whether due to CWF, natural fluoride release from soil minerals, food supplements or tea consumption, especially when the exposure occurs during early brain development.

Fluoride is not an essential nutrient. No physiological function can be defined during the development and growth, for which it is required. Fluoride toxicity is a slow, hidden process. Evolving evidence should inspire scientists and health authorities to re-evaluate claims about the safety of fluoride, especially for the fetus and infant for whom it has no benefit at all.

Of all sources of fluoride, artificially fluoridated water is the most available practical source to eliminate fluoride intake to reduce its human hazards. Our review explains that fluoride could evoke unexpected epidemics in the future.

**Author Contributions:** Both authors are responsible for the intellectual content, literature review, preparation of figures and tables, and drafting of the manuscript. Both authors prepared and approved the final manuscript. Both authors participated in the introduction of references into the text and preparation and improvement of a list of references. All authors have read and agreed to the published version of the manuscript.

**Funding:** This work was undertaken without funding.

**Acknowledgments:** We appreciate the excellent cooperation of Hana Kružíková in the figures' preparation.

**Conflicts of Interest:** The authors do not have any conflict of interest to declare.

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
