# Peer review of "Mechanisms of Fluoride Toxicity: From Enzymes to Underlying Integrative Networks"

_applsci, doi:10.3390/app10207100_

Round 1
Reviewer 1 Report
The review of Strunecka et Strunecky talks about the impact and the mechanisms of fluoride toxicity. This discussion is essential since it will influence future decisions regarding the impact of fluoride toxicity around the world.
This is a high quality and potentially interesting review. The review is well organized and written but certain concerns should be addressed regarding the figures. I would suggest to the authors to improve the quality of the figures.
Author Response
Thank you very much for your review of our manuscript. We improved Figures 3 and 5, where the writings present were too small and blurred.
Reviewer 2 Report
This review article deals with the toxicity of F.
The manuscript was well made and constructed.
It seems that all the "Figures" are produced in higher resolution and should be attached.
Line 50. Please add fluoride "compounds" as for example.
Line 48 - 55. How about describing or adding some information about the effect of F on plants and ecosystems? Of course, this paper deals with the effects on the human body, but it would be good to mention the effects on non-human ecosystems in the introduction.
Line 271-272. Among the mechanisms in which ROS is generated by F, the process of disturbing the flow of electrons should be added. Even in the basic metabolic process, some ROS is produced, and it would be good to mention the effect of F at that time.
Author Response
Thank you very much for your revision and comments.
We revised line 49-59
Fluorine atom does not exist in nature in a free and unmixed state. The mineral fluorite (CaF2) is the main source of fluorine for commercial use. Fluorine readily reacts with other elements to form fluoride compounds, within which it always adopts an oxidation state of −1. Inorganic compounds are formed with hydrogen, metals, and nonmetals. Fluoride, therefore, refers to the negatively charged ion of the fluorine atom. In general, soil fluoride is not available to plants. There are exceptions such as tea plants that are natural accumulators of fluoride from fertilizers [1,2,4]. Fruits such as blueberry and grape concentrate fluoride from soil and it might appear in commercial juices. Fluorine excess had been found detrimental also to plants. Injury to plants is common all-around industries making aluminum, fertilizers, or glass. Special plants might produce toxic organofluorine compound fluoroacetate, which is used as a defense against grazing by herbivores.
Line 282–293
Under normal physiologic conditions, the glutathione (GSH) redox equilibrium regulates a pleiotropic range of functions that includes reactive oxygen species (ROS) scavenging, maintaining cell membrane integrity, signal transduction, and apoptosis. Intracellular oxidative stress may have functional consequences such as increased mitochondrial superoxide production and a chronic inflammatory response. Excessive intracellular oxidative stress may contribute to several body dysfunctions like diabetes, neurological diseases, cancer, and rapid aging. The imbalance between oxidants and antioxidants is regulated by the GSH redox system in the human body. Some authors suggest that oxidative stress is the key mechanism of fluoride toxicity [95,97-99]. Fluoride causes the increased generation of ROS such as superoxide anion, reactive hydroperoxides (ROOH), and hydrogen peroxide (H2O2) (Figure 4). Fluoride exposure can reduce the cellular level of GSH, often resulting in excessive production of ROS [97-99]. Fluoride inhibits the activity of antioxidant enzymes such as catalase, glutathione peroxidase (GPx), GSH reductase (GR), and superoxide dismutase (SOD) (see Table 1).
Reviewer 3 Report
This review addresses the interesting and important topic of fluoride toxicity. In my opinion there is plenty of information that may help scientists and investigators to better understand this issue.
Overall this paper would need an English polishing and minor corrections. Here below are listed some of them along with few comments:
line 18: "we suggest focus" should be changed with "we suggest to focus"
line 33: delete "Dr." and change "the head" with "head"
line 36: change "in one" with "per"
line 49: change "unites" with "reacts"
line 64: in the abstract, in line 14, it is reported "May 2020", while here the Authors cite "June 2020". Which of them is correct? Also, delete "from"
lines 65-66: change "..fluoride, along with search terms enzyme activity..." with "...fluoride, enzyme activity..."
line 76: change "chapter" with "review"
line 82: is 0.01% by weight? Is it intended as 0.01 g of F per 100 g of water?
line 95: are you sure that it is "MgF3" and not "MgF2"?
line 111: "synthesis" of what? In this sentence it seems that the conclusion is missing
line 112: change "of" with "on how"
line 121: change "motive" with "driving"
lines 135-136: the sentence "The NKA ....neuropeptides" does not refer to the fluoride issue. It should be deleted.
lines 171-174: the tables 1 and 2 are very good
line 182: the last part of the sentence looks unclear and probably should be rewritten
line 196: change "for bound" with "to bind"
line 214: change "in millimolar fluoride" with "in millimolar fluoride concentrations"
line 224-225: Figure 3 (right part) is difficult to read, writings are too small and blurred
line 249: delete "of"
lines 288-289: this sentence should be rewritten
lines 291-293: does any sure link exist between fluoride and autism? If not, delete these sentences, otherwise these sentences are unclear: they do not make the point
line 369-370: like Figure 3, the writings present in Figure 5 are too small and blurred
lines 378-387: is there any link between these sentences and the fluoride intake? This link should be, eventually, specified
line 393: change "can been" with "can be"
line 485: change "provides" with "is provided by"
line 607: delete "the".
Author Response
Thank you very much for your comments and kind corrections of English language and style. We accepted all but one as indicated in our manuscript.
line 82 we revised as Fluoride in 0.01% concentration
line 95: are you sure that it is "MgF3" and not "MgF2"?
It is OK. trifluoromagnesate (MgF(3)(-)that is a transition state analogue for phosphoryl moiety PO(3)(-)transfer.
line 111: "synthesis" of what? In this sentence it seems that the conclusion is missing. We deleted the sentence The removal of PPi by PPase drives these reactions in the direction of synthesis.
line 224-225: Figure 3 (right part) is difficult to read, writings are too small and blurred. Thank you for your comment, we improved figures 3 and 5
line 288-299
have been rewritten:
It has been found that a high amount of fluoride can trigger oxidative stress by sabotaging the intracellular anti-oxidative defense system. The levels of ROS were significantly increased, while the activity of SOD was decreased in the blood of subjects from endemic fluorosis area in China [5].
lines 291-293: does any sure link exist between fluoride and autism? If not, delete these sentences, otherwise these sentences are unclear: they do not make the point
We insist on this postulation based on given references and our previous papers
lines 378-387: is there any link between these sentences and the fluoride intake? This link should be, eventually, specified
It is a link between melatonin – fluoride, and ASD